# Bone-in-culture array as a platform to model early-stage bone metastases and discover anti-metastasis therapies

Hai Wang[1,2,3,4], Lin Tian[1,2,3,5], Amit Goldstein[1,2,3], Jun Liu[1,2,3], Hin-Ching Lo[1,2,3,6], Kuanwei Sheng[6,7], Thomas Welte[1,2,3], Stephen T.C. Wong[8,9], Zbigniew Gugala[10], Fabio Stossi[2,3], Chenghang Zong[2,6,11], Zonghai Li[4], Michael A. Mancini[2,3] & Xiang H.-F. Zhang[1,2,3,11]

The majority of breast cancer models for drug discovery are based on orthotopic or subcutaneous tumours. Therapeutic responses of metastases, especially microscopic metastases, are likely to differ from these tumours due to distinct cancer-microenvironment crosstalk in distant organs. Here, to recapitulate such differences, we established an *ex vivo* bone metastasis model, termed bone-in-culture array or BICA, by fragmenting mouse bones preloaded with breast cancer cells via intra-iliac artery injection. Cancer cells in BICA maintain features of *in vivo* bone micrometastases regarding the microenvironmental niche, gene expression profile, metastatic growth kinetics and therapeutic responses. Through a proof-of-principle drug screening using BICA, we found that danusertib, an inhibitor of the Aurora kinase family, preferentially inhibits bone micrometastases. In contrast, certain histone methyltransferase inhibitors stimulate metastatic outgrowth of indolent cancer cells, specifically in the bone. Thus, BICA can be used to investigate mechanisms involved in bone colonization and to rapidly test drug efficacies on bone micrometastases.

[1] Lester and Sue Smith Breast Center, Baylor College of Medicine, One Baylor Plaza, Houston, Texas 77030, USA. [2] Dan L. Duncan Cancer Center, Baylor College of Medicine, One Baylor Plaza, Houston, Texas 77030, USA. [3] Department of Molecular and Cellular Biology, Baylor College of Medicine, One Baylor Plaza, Houston, Texas 77030, USA. [4] State Key Laboratory of Oncogenes and Related Genes, Shanghai Cancer Institute, Renji Hospital, Shanghai Jiao Tong University School of Medicine, Shanghai 200032, China. [5] Verna & Marrs McLean Department of Biochemistry and Molecular Biology, Baylor College of Medicine, One Baylor Plaza, Houston, Texas 77030, USA. [6] Department of Molecular and Human Genetics, One Baylor Plaza, Houston, Texas 77030, USA. [7] Graduate Program in Integrative Molecular and Biomedical Sciences, Baylor College of Medicine, One Baylor Plaza, Houston, Texas 77030, USA. [8] Department of Systems Medicine and Bioengineering, Houston Methodist Research Institute, Weill Cornell Medical College, 6670 Bertner Avenue, Houston, Texas 77030, USA. [9] Department of Radiology, Houston Methodist Hospital, 6670 Bertner Avenue, Houston, Texas 77030, USA. [10] Department of Orthopaedic Surgery and Rehabilitation, University of Texas Medical Branch, 301 University Boulevard, Galveston, Texas 77555, USA. [11] McNair Medical Institute, Baylor College of Medicine, One Baylor Plaza, Houston, Texas 77030, USA. Correspondence and requests for materials should be addressed to X.H.-F.Z. (email: xiangz@bcm.edu).

In the clinic, primary breast tumours are usually surgically removed soon after diagnosis, leaving patients 'tumour-free'. However, 20–40% of breast cancer survivors will eventually suffer metastasis to distant organs, sometimes years after surgery[1,2]. Thus, the life-threatening enemy is typically not the bulk of primary tumours, but the dispersed metastatic seeds, which have disseminated to distant organs, may be temporarily dormant, and may resume aggressive outgrowth under certain yet-to-be-identified conditions. Current adjuvant therapies are intended to eliminate these cells. However, the therapeutic decisions and strategies are usually based on pathological features of primary tumours. Metastases are likely to differ from their parental primary tumours due to Darwinian selection and/or adaptation in a different milieu. In either case, the microenvironment in distant organs plays a critical role in driving the selection and/or adaptation of cancer cells.

Bone is the organ most frequently affected by breast cancer metastasis[3–7]. Its diagnosis relies on skeletal-related events, including pathological fractures[8]. Mechanistically, these events are caused by the vicious cycle between osteoclasts and cancer cells[4,9]. Cancer cells can release factors such as parathyroid hormone-related protein, which will stimulate the production of Receptor activator of nuclear factor kappa-B ligand (RANKL) by osteoblasts, leading to the activation of osteoclasts[9–11]. Reciprocally, growth factors such as insulin-like growth factor-1 and transforming growth factor-β are released from dissolved bone matrix to further fuel cancer cell growth[9,12].

We have recently provided evidence supporting a pre-osteolytic phase of bone colonization before the vicious cycle[13]. In this phase, breast cancer cells, especially the luminal subtype, tightly interact with cells in the osteoblast lineage, or osteogenic cells. Osteoclasts, on the other hand, do not appear to be involved until the transition from 'osteogenic' lesions to 'osteolytic' lesions. Consistent with this finding, cancer cells injected through the iliac artery soon became tightly embedded in bone tissues and could only be dissociated with protease digestion, even after bone fragmentation[14]. This characteristic led us to establish an *ex vivo* model named 'bone-in-culture array' or BICA. Here we provide evidence demonstrating that BICA mimics cancer–bone interactions in the pre-osteolytic phase, and also recapitulates transitions to the osteolytic phase. Thus, it represents a preclinical platform that may fill the gap between *in vitro* and *in vivo* models, and accelerate mechanistic and pharmacological studies of bone metastasis.

## Results

**BICA provides a bone-like microenvironment**. BICA is based on a technique that we have previously established, namely intra-iliac artery (IIA) injection[13], which selectively delivers cancer cells into the hindlimbs of mice through arterial circulation. After injection, cancer cells usually home to the spongy bone of the tibia or femur bones. To develop BICA, we extracted and fragmented epiphysis and metaphysis of hindlimb bones containing the cancer cells (Fig. 1a and Supplementary Fig. 1a). The bone fragments (0.5–1.5 mm in diameter and 0.2–0.4 g cm$^{-3}$ in mineral density; Supplementary Fig. 1b,c) can be maintained in tissue culture for up to 6 weeks without significant loss of viability (Supplementary Fig. 1d). Since the breast cancer cells utilized in this study are engineered to express luciferase, bioluminescence imaging can be used to quantify viable cancer cells. Cancer cells remain confined within fragments during this time, probably due to the tight interaction between cancer and bone cells (Fig. 1a). About 20–50 bone fragments can be obtained from one mouse, thus greatly reducing the number of mice needed for each experiment and making multiple parallel applications feasible. In the following paragraphs of this section, we describe several experiments performed to compare BICA with *in vivo* bone lesions (IVBL) introduced by IIA injection. In these comparisons, we also included orthotopic tumours and cancer cells maintained in two-dimensional (2D) cultures to represent the non-bone microenvironment.

To determine whether the microenvironment of cancer cells in BICA and IVBL are similar, we performed immunofluorescent staining of alkaline phosphatase (ALP), collagen I (Col-I) and cathepsin K (CTSK). ALP and Col-I are expressed in the cells of the osteoblast lineage, whereas CTSK is a marker of activated osteoclasts. The expression pattern of these molecules is very similar in the two models (Fig. 1b and Supplementary Fig. 1e), and consistent with what we have previously found in bone micrometastases at the pre-osteolytic stage[13].

Peri-vascular niche has been increasingly implicated in regulating cellular fates of disseminated tumour cells. We examined whether endothelial cells remain in BICA. Indeed, immunofluorescence staining of CD31$^+$ cells uncovered that at least some of these cells can persist for up to 3 weeks in BICA (Supplementary Fig. 1f), providing potential opportunities to study the interaction between cancer cells and the peri-vascular niche.

Although activated osteoclasts are absent at early time points, we noticed that some monocytes, the precursors of osteoclasts, remain close to bone fragments in BICA (Fig. 1c). To test whether these monocytes still maintain the potential to differentiate into osteoclasts, we added macrophage colony-stimulating factor (M-CSF) and RANKL to the medium and performed staining of tartrate-resistant acid phosphatase (TRAP) to examine osteoclastic activities. Indeed, M-CSF/RANKL treatment induced TRAP$^+$, multinuclear osteoclasts (Supplementary Fig. 2a) indicating that monocytes in BICA retain differentiation abilities. To ask whether monocytes can spontaneously differentiate into activated osteoclasts in BICA, we applied cancer cell models that rapidly induce osteolytic vicious cycle *in vivo*, namely MDA-MB-231 cells and one of its osteotropic subpopulations, SCP28 cells[15]. At relatively late time points (3–5 weeks), we examined osteoclast differentiation by CTSK or TRAP staining. As expected, these cancer cells induced spontaneous osteoclastogenesis in BICA, both to monocytes suspended in medium (Supplementary Fig. 2b) and to those remaining in bone fragments (Fig. 1d and Supplementary Fig. 2c). In contrast, cancer cells that undergo indolent metastatic growth and prolonged pre-osteolytic stage (MCF-7) did not exhibit the same activity (Supplementary Fig. 2b,c). The activated osteoclasts in BICA appear to dissolve bone matrix as evidenced by an increased surface-to-volume ratio in SCP28-containing bone fragments (Supplementary Fig. 2d). Moreover, SCP28 cells showed a similar growth kinetic to *in vivo* condition and grew more rapidly as compared to the parental MDA-MB-231 cells in early time points, confirming that BICA can recapitulate metastasis bone tropism (Supplementary Fig. 2e). The growth kinetics of SCP28 cells in BICA closely mimicked the same cells in IVBL till 4 weeks after injection (Supplementary Fig. 2f), when tumour-induced osteoclastogenesis had already started. However, the growth then became retarded, probably because the tumour burden had saturated the bone surface area in the fragments. Thus, BICA may not be able to recapitulate full-fledged vicious cycle. Despite this limitation, our data demonstrated the suitability of BICA in modelling pre-osteolytic stage and perhaps also the transition from pre-osteolytic to osteolytic stages during bone colonization.

To further characterize BICA at a molecular level, we performed transcriptomic profiling to determine similarities or differences between BICA and IVBL. Cancer-containing bone

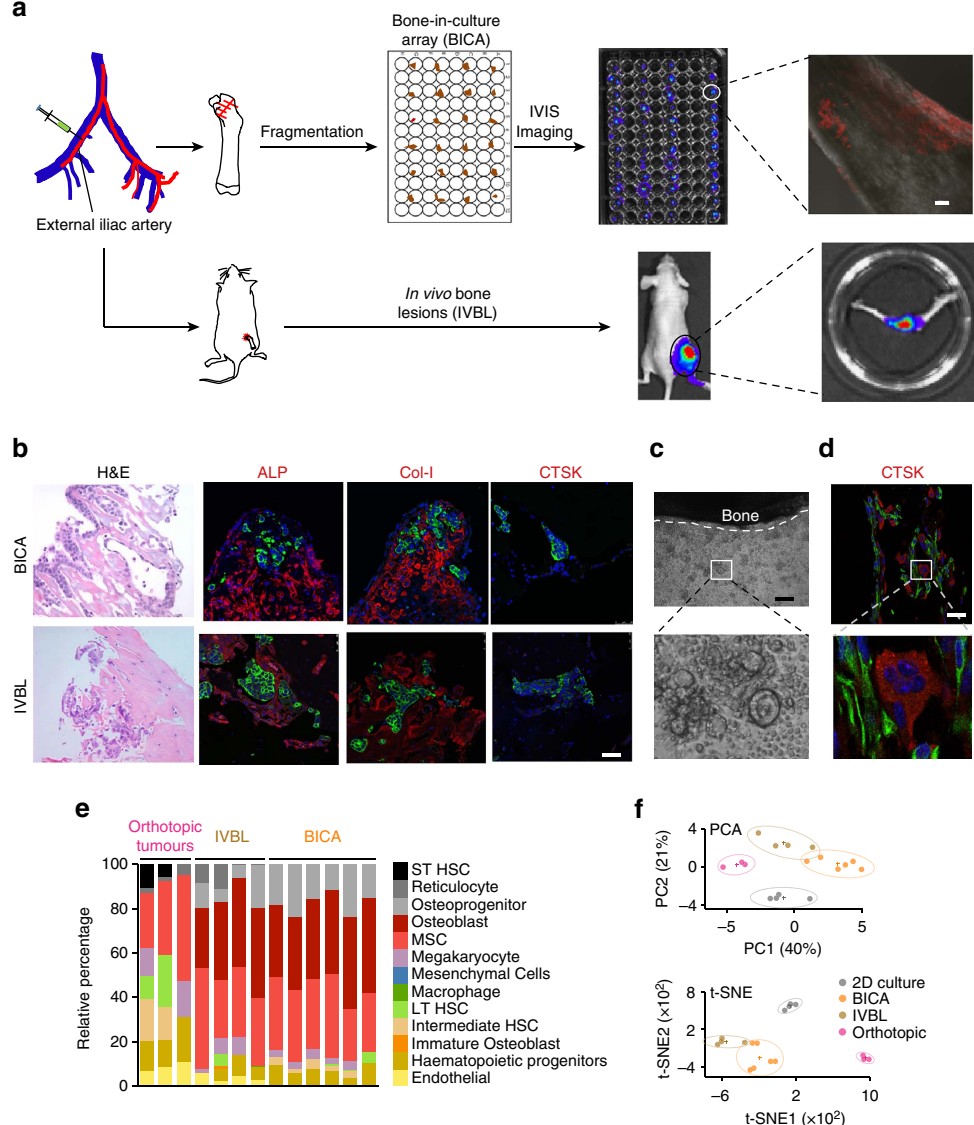

**Figure 1 | BICA provides a bone-like microenvironment.** (**a**) Schematic of IIA injection-based establishment of BICA and IVBL. Luciferase/fluorescence protein-tagged cancer cells are inoculated into the external iliac artery of mice. The injected animals will either be killed for BICA or left alive to give rise to IVBL. For BICA set-up, femur and tibia bones are extracted. Epiphysis and metaphysis are segmented into pieces (1–2 mm in diameter), which will then be arranged into low-attachment 96-well plates with cell culture media. Cancer cells can be traced using confocal microscopy and bioluminescence imaging. Scale bar in upper right panel, 100 μm. (**b**) Haematoxylin and eosin, and fluorescence staining of osteoblast (ALP and Col-I) and osteoclast (CTSK) markers to characterize the microenvironment niche of MCF-7 cells (green) in BICA and IVBL. The quantification is shown in Supplementary Fig. 1d. Scale bar, 25 μm. (**c**) Inverted microscope images of a bone fragment in BICA and haematopoietic cells staying around it. The white dotted line indicates the border of bone fragment. Scale bar, 100 μm. (**d**) Fluorescence staining in BICA indicated CTSK + (red) osteoclasts and human-specific Vimentin + (green) SCP28 cells on a BICA fragment of 5 weeks culture without treatment of M-CSF and RANKL. Scale bar, 25 μm. (**e**) Stromal components of MCF-7 orthotopic tumours, IVBL and BICA predicted by CYBERSORT algorithm to mouse sequences in RNA-seq results. Each column represents one tumour (orthotopic), one bone (IVBL) or pooled bone segments from one animal (BICA). ST HSC, short-term haematopoietic stem cells; LT HSC, long-term haematopoietic stem cells. (**f**) Principle component analysis (PCA) of human sequences in the RNA-seq results of the indicated MCF-7 specimens. Upper panel: PCA based on the top 100 genes with highest average expression across all samples. PC1/2, principle components 1/2. Lower panel: t-Distributed Stochastic Neighbor Embedding (t-SNE) analysis[17].

segments in both BICA and IVBL were subjected to RNA-seq along with 2D culture samples and orthotopic tumours. Since we used MCF-7 cells as a proof of principle, the cancer cell and host transcriptomes are human and mouse, respectively, and can be separated by mapping them to different reference genomes (Supplementary Fig. 3). We first applied CIBERSORT[16] to the mouse sequences to deduce cell types constituting the microenvironments of BICA, IVBL and orthotopic tumours. The major cellular components are comparable between BICA

and IVBL, both of which differ markedly from those in orthotopic tumours. Importantly, among the 13 major cell types included into the analysis, osteoblasts, osteoprogenitors and mesenchymal stem cells together account for over two-thirds of microenvironment cells in both BICA and IVBL, but not in orthotopic tumours (Fig. 1e). These data support our previous finding that the niche of early-stage bone colonization is predominantly osteogenic[13]. Principle component analysis and t-Distributed Stochastic Neighbor Embedding analysis[17] of

human RNAs indicated that the transcriptomic profiles of cancer cells in BICA more closely mimic those in IVBL, as compared to cancer cells in 2D and in orthotopic tumours (Fig. 1f). Taken together, these results provide additional evidence supporting BICA as a platform to mimic the bone microenvironment.

**BICA provides opportunities to study metastasis dormancy**. Some indolent cancer cells undergo delayed growth after being introduced into the bone by IIA injection[13], suggesting a period of dormancy. To track cellular fates of cancer cells in different microenvironment, we used inducible expression of histone protein 2B (H2B) fused with either firefly luciferase (Fluc) or green fluorescent protein (GFP). Similar systems were used before to study quiescent adult stem cells[18]. When the expression of H2B-Fluc/GFP was fully induced in all cells, we withdrew doxycycline and followed GFP or bioluminescence signals. We expected quiescent cells would maintain the expression of the fusion proteins in nucleus. Proliferating cells would have the fusion proteins diluted in each cell, but the overall quantity of the proteins should maintain the same level. Cell death would cause a net loss of these proteins. We validated these predictions in 2D cultures by immunofluorescence staining of GFP in conjunction with Ki67 or BAX to indicate proliferation and apoptosis, respectively (Fig. 2a). We chose BAX as an apoptosis marker here because Cleaved Caspase 3 is absent in MCF-7 (ref. 19). In parallel, we also used constitutive expression of Fluc/GFP to simply follow numbers of viable cancer cells (Fig. 2a). The combination of constitutive and inducible systems allowed us to delineate cancer cell fates: if a cell population is mainly consists of proliferating and dying cells, one would expect to see an alteration of constitutive Fluc/GFP (depending on the balance between proliferation and cell death), but a net decrease of induced H2B-Fluc/GFP signals. On the contrary, dormant cell population would have stable signals in both settings.

When this strategy was applied to MCF-7 cells, we found that constitutive Fluc/GFP exhibited delayed increase in both IVBL and BICA samples for about 2 weeks. In contrast, this signal immediately entered exponential growth in 2D cultures and on introduction to mammary fat pads (Fig. 2b). Cancer cells cultured under several other conditions also failed to mimic the growth kinetics of IVBL, including those growing on plates coated with specific extracellular proteins, in three-dimensional (3D) suspension medium, or directly dropped on top of bone fragments (instead of getting incorporated via IIA injection; Supplementary Fig. 4a–c). In parallel, the induced H2B-Fluc signal of MCF-7 cells was stable for 2 weeks in BICA before starting to decrease, whereas the same signal in 2D cultures rapidly decreased (Fig. 2c). Immunofluorescence staining confirmed that cancer cells in BICA maintained expression of H2B-GFP and were negative for both Ki67 and BAX (Fig. 2d). Since endothelial cells persist in BICA (Supplementary Fig. 1f), we asked whether dormant cancer cells localize to peri-vascular niche[20,21]. Indeed, as reported in the previous literature[21], we detected indolent MDA-MB-231 cells adjacent to CD31$^+$ cells (Supplementary Fig. 4d), supporting the significance of this niche. Taken together, these data strongly suggest that some indolent cancer cells undergo a short period of dormancy in BICA, providing an opportunity to study the underlying mechanisms.

**BICA recapitulates the osteogenic niche**. We asked whether the cancer–niche interaction in BICA mimics that of IVBL. Analyses of the RNA-seq results revealed that cancer cells in both BICA and IVBL exhibited increased expression of mTOR target genes as well as genes involved in cell–cell adhesions (Fig. 3a). This is consistent with our previous studies showing that luminal breast

cancer cells utilize E-cadherin to form heterotypic adherens junctions with N-cadherin expressed by osteogenic cells[13]. This interaction activates the mTOR pathway in cancer cells and drives metastatic progression during early-stage bone colonization[13] (Fig. 3b). In the following experiments, we asked whether BICA could recapitulate the cancer–niche interaction observed in whole animals by pharmacological or genetic perturbations of the heterotypic adherens junctions–mTOR pathway in both BICA and IVBL.

First, we conditionally knocked out N-cadherin in osteogenic cells or endothelial/haematopoietic cells using *Osterix-cre* or *Tie2-cre* alleles in combination with *Cdh2$^{flox}$* (encoding N-Cadherin) alleles, respectively (Fig. 3c and Supplementary Fig. 5a). Bone colonization of syngeneic AT3 cells was significantly impaired in *Osterix-cre;Cdh2$^{f/f}$* animals, as compared to control animals and *Tie2-cre;Cdh2$^{f/f}$* (Fig. 3d and Supplementary Fig. 5b). A similar reduction in tumour growth in BICA was also observed using these mouse strains as sources for bone fragments (Fig. 3e and Supplementary Fig. 5c), suggesting that N-cadherin is important for cancer colonization both in IVBL and in BICA.

Second, we tested the efficacies of an mTOR inhibitor (Torin 1), a mitogen-activated protein kinase (MAPK) inhibitor (PD98059) and a neutralizing antibody against E-cadherin on cancer cells in BICA, IVBL or 2D cultures. Both the mTOR inhibitor and anti-E-cadherin treatment significantly inhibited the progression of IVBL (Fig. 3f,g), as expected based on our previous findings (Fig. 3b)[13]. On the other hand, the MAPK inhibitor failed to affect bone colonization (Fig. 3h). When applied to BICA, both the mTOR inhibitor and anti-E-cadherin treatment could similarly inhibit cancer cells with high efficiencies (Fig. 3i,j). Importantly, the efficacy of anti-E-cadherin treatment is higher in BICA as compared to 2D cultures (Fig. 3j), indicating that BICA enhances sensitivity of cancer cells to E-cadherin blockade. In contrast, BICA confers resistance to the MAPK inhibitor, a phenomenon also seen in IVBL (Fig. 3h) but not in 2D culture (Fig. 3k).

Thus, BICA, similar to IVBL but different from 2D cultures, exhibited expected responses to various molecular perturbations based on our understanding of the osteogenic niche. These data support the reliability of BICA as a model to recapitulate cancer–niche interaction in the bone.

**Multiple parallel drug tests using BICA**. As a proof of principle, we chose a small library of chemical compounds and tested their effects on cancer cells in BICA. The library contains 68 small-molecule epigenetic modulators (designated as EG library, available from Selleckchem, Catalogue # L1900). This was chosen based on the rationale that adaptation of cancer cells to the bone microenvironment may involve epigenomic reprogramming, which might be modulated by some of these compounds. For comparison, we simultaneously applied these compounds to the same cancer cells in 2D cultures. We intended to identify compounds exhibiting differential effects in BICA as compared to 2D cultures.

Two rounds of tests were conducted. In the first round, the EG library was divided into functionally related groups containing five to six compounds (Supplementary Table 1). Groups that exhibited strong effects were then separated and subjected to the second-round tests (Fig. 4a). In the first round of tests, diverse effects of different groups were observed in both BICA and 2D cultures (Fig. 4b). In particular, we noticed that Aurora kinase inhibitors and histone methyltransferase (HMT) inhibitors exhibited significant but opposite effects: the former inhibited and the latter stimulated tumour growth in BICA, as compared to the same cancer cells in 2D cultures (Fig. 4b). Histone

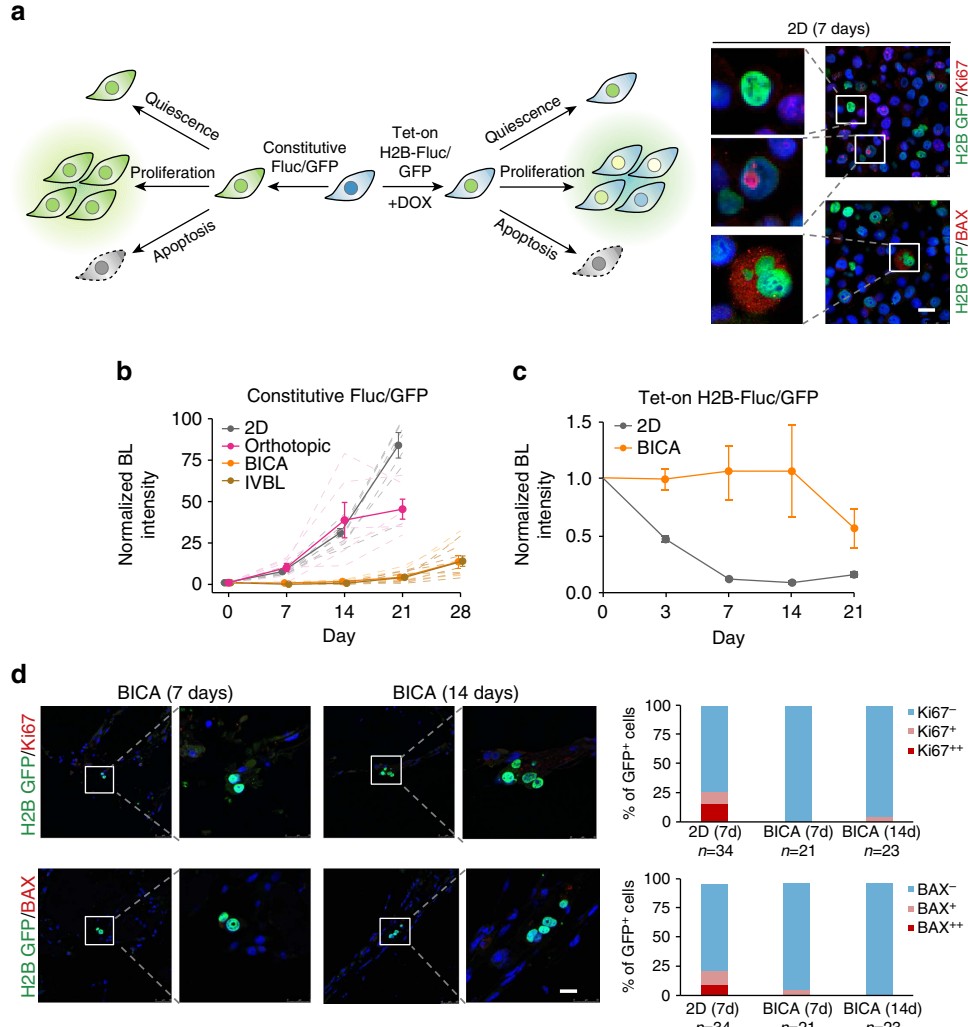

**Figure 2 | Dormant cancer cells in BICA.** (**a**) Experimental systems to determine cancer cell fates. Left: schematics of experimental design combining constitutive expression of Fluc/GFP and inducible expression of H2B fused with Fluc or GFP (Tet-on-H2B-Fluc/GFP). Three different cellular fates and the predicted Fluc/GFP signal statuses are indicated in both cases. Right: representative immunofluorescent pictures are shown to validate the inducible H2B-Fluc/GFP system. GFP (green) was immune-stained in conjunction with either Ki67 or BAX (red). Scale bar, 25 μm. (**b**) Kinetics of bioluminescence signals of MCF-7 cells that constitutively express Fluc/GFP in 2D cultures (8 wells as technical replicates), orthotopic tumours (mammary fat pad injection of $1 \times 10^6$ cells, $n = 6$ animals), BICA (IIA injection of $5 \times 10^5$ cells, $n = 9$ fragments) and IVBL (IIA injection of $5 \times 10^5$ cells, $n = 8$ athymic nude mice). Cancer cells are quantified by bioluminescence (BL) intensity. All data are normalized to day 0 values right after injection or seeding. Thick lines represent group average and thin/dot lines represent individual animals (orthotopic and IVBL), bone fragment (BICA) or tissue culture plates (2D). Error bars: s.e.m. The experiment was replicated twice in the laboratory with consistent results and a representative one is shown. (**c**) Kinetics of bioluminescence signals of MCF-7 cells that express inducible H2B-Fluc/GFP in 2D cultures (12 wells as technical replicates) and BICA (IIA injection of $5 \times 10^5$ cells, $n = 6$ fragments). All data are normalized to day 0 values right after injection or seeding. Error bars: s.e.m. The experiment was replicated twice in the laboratory with consistent results and a representative one is shown. (**d**) Immunofluorescence staining and quantification of GFP (green) in conjunction with Ki67 (red) or BAX(red) on MCF-7 cells in BICA 2 weeks after IIA injection. $N = GFP^+$ cancer cells observed in each group. Scale bar, 25 μm.

deacetylase inhibitors also exhibited strong anti-tumour effects in both 2D cultures and BICA. However, a toxicity test using tumour-free bone fragments revealed strong negative impacts of these compounds on the viability of healthy bone cells (Supplementary Fig. 6). Therefore, we only focused on the Aurora kinase inhibitors and HMT inhibitors. The second round of tests identified danusertib as an individual drug responsible for the inhibitory effects seen for the Aurora kinase inhibitors (Fig. 4c), and EPZ-6438 and MM-102 for the stimulatory effects of the HMT inhibitor group (Fig. 4d). These drugs did not affect the viability of tumour-free bone segments when used at the same dosage (Supplementary Fig. 6). Therefore, we chose to focus on danusertib, EPZ-6438 and MM-102 for further analyses.

**Danusertib preferentially eliminated cancer cells in bone.** Aurora kinases play an important role in regulation of mitosis and cell proliferation[22]. They have recently been implicated in epigenetic modification of histones. Danusertib is a pan-Aurora kinase inhibitor[23], and has been tested in phase 2 clinical trials[24]. Here our results pointed to an increased efficacy of danusertib on cancer cells in BICA as compared to those in 2D cultures (Fig. 5a). Specifically, 100 nM danusertib could achieve over 90% inhibition rate of tumour growth in BICA, but only ~60% in 2D or 3D cultures (Fig. 5a and Supplementary Fig. 7a,b). This difference is unlikely due to different levels of baseline proliferation, as the proliferation rate in 3D suspension medium is much lower than that in 2D or BICA, yet the inhibition rate of

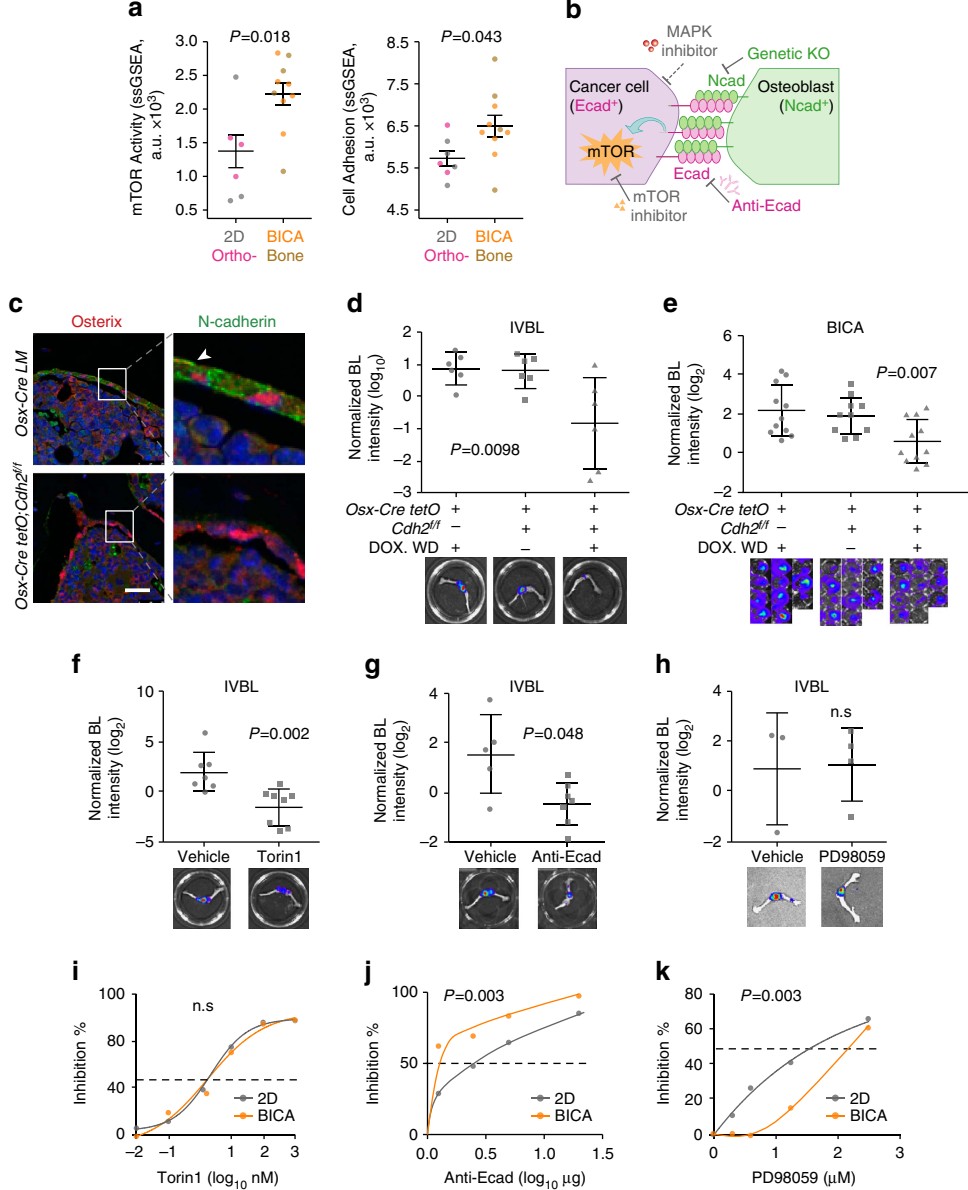

**Figure 3 | Cancer cells in BICA recapitulate the cancer–niche interaction. (a)** ssGSEA analyses of RNA-seq data identified coordinated alterations of mTOR target genes (left) and cell adhesion pathway genes (right) of MCF-7. Each dot represents an individual animal (orthotopic (Ortho-) and IVBL), pooled bone fragments from an animal (BICA) or a tissue culture plate (2D). Error bars: s.e.m. $P$ value was determined by two-tailed unpaired Mann–Whitney $U$-test. **(b)** Schematic shows the previously demonstrated mechanism of cancer–niche interaction. Various perturbations are applied in subsequent figure panels as indicated, including an mTOR inhibitor, an antibody against E-cadherin (anti-Ecad) and genetic knockout (KO) of N-cadherin (Ncad). As a comparison, a MAPK inhibitor is also used. **(c)** Representative pictures demonstrating the conditional knockout of N-cadherin in Osterix+ cells in the bone of *Osx-cre tetO; cdh2f/f* mice or the litter mate (LM) control with only *Osterix-cre*. Red: Osterix; green: N-cadherin. Scale bar, 50 μm. **(d,e)** Quantification of tumour burdens in IVBL in host mice with inducible and conditional KO of N-cadherin in osteogenic cells (Osterix+ and descendent cells). Syngeneic AT3 **(d)** or MCF-7 **(e)** cancer cells were injected via IIA. KO is induced by withdrawal (WD) of doxycycline (Dox). Each dot represents an animal (IVBL, day 14) **(d)** or a bone fragment (BICA, day 21) **(e)**. $P$ values are determined by one-way analysis of variance (ANOVA). Error bars: s.d. The experiment was replicated twice and showed consistent results. The combined results are shown. **(f–h)** The effects of Torin 1 **(f)**, anti-Ecad **(g)** and PD98059 **(h)** on IVBL. In all, $5 \times 10^5$ MCF-7 cells were IIA-injected and samples were collected on day 21. Metastatic burden was quantified by bioluminescence (BL) intensity and normalized to day 0. $P$ values were determined by two-tailed unpaired Mann–Whitney $U$-test. **(i–k)** Dose–response curves of Torin 1 **(i)**, anti-Ecad **(j)** and PD98059 **(k)** on MCF-7 cells in BICA or 2D cultures. For 2D cultures, six technical replicates were included for each drug concentration. For BICA, $N = 6$ bone fragments for each drug concentration. Samples are under corresponding treatment for 19 days and measured by BL. $P$ values are determined by repeated measures ANOVA tests.

danusertib in 3D culture is similar to that in 2D culture, but lower than that in BICA (Supplementary Fig. 7b). Thus, the bone microenvironment appears to enhance sensitivity of cancer cells to danusertib.

We sought to validate this increased sensitivity in additional cancer models. To this end we examined MDA-MB-361, another cancer cell line that can slowly colonize bone. We also tested a patient-derived xenograft (PDX) model using BICA. In both

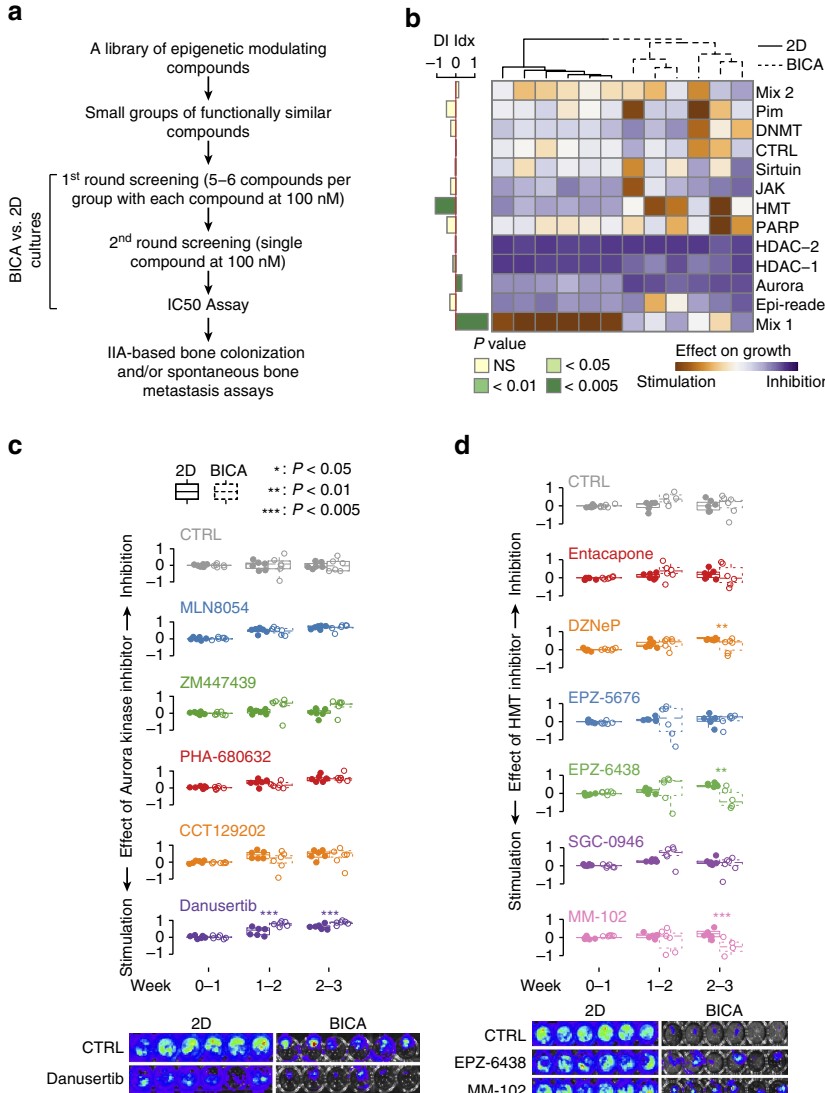

**Figure 4 | Parallel tests of a collection of epigenomic compounds using BICA as a preclinical platform.** (**a**) Schematic shows the major steps of parallel 2D/BICA tests to identify compounds with impacts on cancer cells in the bone microenvironment. (**b**) A heat map summarizing the output of first-round testing using groups of functionally related compounds. (See Supplementary Table 1 for detailed information on each group.) MCF-7 samples are under corresponding treatment for 3 weeks and measured by bioluminescence. The differential inhibition (DI) index is obtained by subtracting the 2D inhibition score from BICA inhibition score (BICA − 2D). The positive value means the drug works better in BICA in terms of growth inhibition (for example, Aurora); the negative value means the drug inhibits cancer cell growth in 2D condition better (for example, HMT). Each column represents an independent cell culture (2D) or bone fragment (BICA). The $P$ values are determined by two-tailed unpaired Mann–Whitney $U$-test. (**c**) Results of the second-round test on Aurora kinases inhibitors. Samples are under corresponding treatment for 3 weeks and measured by bioluminescence. CTRL, control. $P$ value is determined by two-tailed unpaired Mann–Whitney $U$-test with multiple-test correction. $N = 6$ bone fragments in each group. Representative bioluminescence images are shown at the bottom of the panel. (**d**) Results of the second-round test on HMT inhibitors. Samples are under corresponding treatment for 3 weeks and measured by bioluminescence. $P$ value is determined by two-tailed unpaired Mann–Whitney $U$-test with multiple-test correction. $N = 6$ bone fragments in each group. Representative bioluminescence images are shown at the bottom of the panel.

cases, danusertib exhibited similar inhibitory effects as with MCF-7 cells (Fig. 5b,c).

We next tested the efficacy of danusertib *in vivo*. We were especially interested in examining the effects of danusertib on early-stage, microscopic bone metastases. Therefore, we chose to use indolent models with a prolonged pre-osteolytic stage. The bone colonization kinetics of MCF-7 cells were characterized in our previous studies. The osteolytic vicious cycle typically occurs over 5 weeks after IIA injection[13], providing a time window to examine drug effects in the pre-osteolytic stage. Treatment with danusertib almost completely abolished the progression of pre-osteolytic bone lesions (Fig. 5d and Supplementary Fig. 7c).

The same dosage of danusertib also reduced orthotopic tumour growth, although to a lesser degree as compared to the reduction of bone colonization (Supplementary Fig. 7d).

To further validate the efficacy of danusertib on spontaneous bone metastases, we used 4T1.2 cell line as a model[25]. When transplanted into the mammary glands of syngeneic Balb/c mice, 4T1.2 cells can give rise to orthotopic tumours that spontaneously disseminate to other organs, including bone. We removed the orthotopic tumours when they reached $0.5\,cm^3$ and monitored metastases to various organs. Danusertib treatment was initiated after orthotopic tumour removal. The 'adjuvant' danusertib treatment significantly reduced spontaneous bone metastases

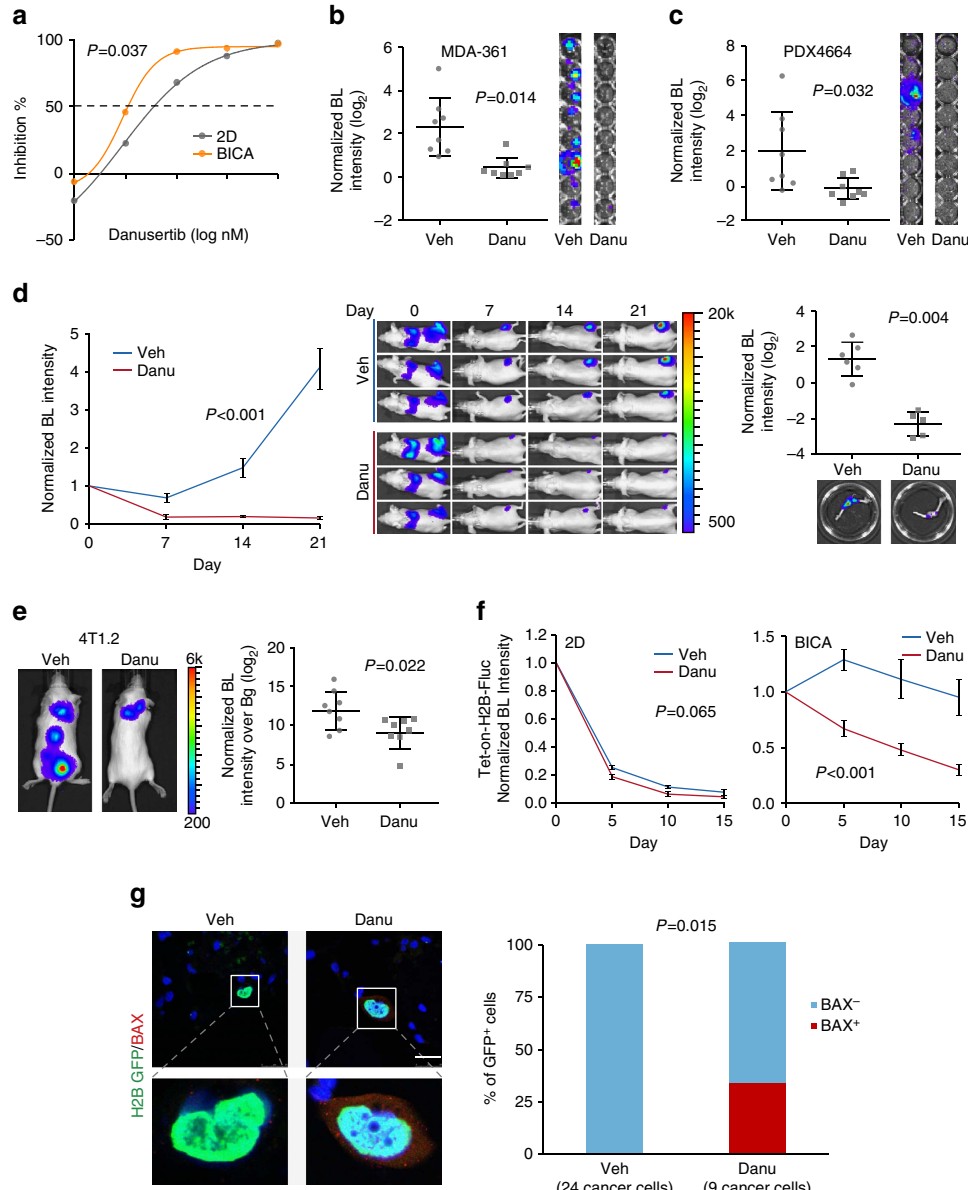

**Figure 5 | Danusertib preferentially eliminated cancer cells in the bone. (a)** Dose responsive curves of MCF-7 cells to danusertib in 2D culture or BICA. For 2D cultures, six technical replicates were included for each drug concentration. For BICA, $N = 6$ bone fragments for each drug concentration. Samples are under corresponding treatment for 3 weeks and measured by bioluminescence (BL). $P$ values are determined by repeated measures analysis of variance (ANOVA) tests. **(b,c)** The effects of danusertib (100 nM) on MDA-MB-361 cells **(b)** and PDX4664 cells **(c)** in BICA. $N = 8$ bone fragments in each group. Samples are under corresponding treatment for 3 weeks. Veh, vehicle; Danu, danusertib. $P$ values are determined by Student's $t$-test (two-tailed) with Welch correction. Error bars: s.d. **(d)** The effects of danusertib on IIA-injected MCF-7 bone lesions. Left: growth curves as measured by *in vivo* BL imaging. $N = 6$ and 5 athymic nude mice for the two groups, respectively. $P$ values are determined by repeated measures ANOVA tests on the growth curves. Error bars: s.e.m. Middle: representative BL pictures show bone lesion arising with or without danusertib treatment. Right: quantification of BL intensity after hindlimbs are extracted. $P$ values were determined by two-tailed unpaired Mann–Whitney $U$-test. Error bars: s.d. **(e)** The effects of danusertib on spontaneous bone metastasis of 4T1.2 model. The treatment of danusertib started after orthotopic tumour resection. Metastasis was quantified 12 days after the surgery, as shown in the dot plot. $N = 8$ Balb/c mice in each experimental group. Bg, background. The $P$ value was determined by $t$-test (two-tailed). Error bars: s.d. **(f)** The effects of danusertib on dormant MCF-7 cells in BICA as measured by Tet-on inducible H2B-Fluc/GFP system. $N = 6$ and 8 bone fragments in each group, respectively. $P$ values are determined by repeated measures ANOVA tests. Error bars: s.e.m. **(g)** Immunofluorescence staining and quantification of GFP (green) in conjunction with BAX (red) on MCF-7 cells in BICA with 1 week of treatment after IIA injection. $N = 8$ bone fragments in each group. Numbers in parentheses indicate the quantity of cancer cells examined. Scale bar, 25 μm. The $P$ value was determined by Fisher's exact test.

(Fig. 5e), but not local recurrences of orthotopic tumours (Supplementary Fig. 7e).

Finally, we asked whether danusertib inhibits bone colonization by eliminating dormant cancer cells, which would be highly desirable in the clinic to permanently decrease risks of

recurrence. Towards this end, we used the inducible H2B-Fluc/GFP system and found that danusertib decreased the otherwise stable H2B-Fluc/GFP signals in BICA, a strong indication of reduction of dormant cancer cells (Fig. 5f and Supplementary Fig. 7f). This was supported by the BAX

expression observed in H2B-GFP⁺ cells in danusertib-treated bone fragments (Fig. 5g).

Taken together, these data support that danusertib treatment represents an effective strategy to prevent bone recurrences, possibly by eliminating dormant disseminated tumour cells in the bone.

**EPZ-6438 and MM-102 stimulate tumour progression in bone.** EPZ-6438 targets EZH2 (ref. 26), and MM-102 is a potent WDR5/MLL interaction inhibitor[27]. Both of these compounds stimulated tumour growth in BICA. Moreover, this stimulatory effect was BICA-specific (Fig. 4d and Supplementary Fig. 7b). In 2D cultures, these two compounds had modest inhibitory effects on cancer cell viability at a dosage of 10 μM (35% inhibition for EPZ-6438 and 13% inhibition for MM-102). However, a clear dose-dependent increase of tumour burden was seen when applied to BICA (Fig. 6a). The same effects of these drugs were

also seen using MDA-MB-361 as a second model in BICA (Fig. 6b). We then examined whether EPZ-6438 (chosen over MM-102 because it has been tested *in vivo*[28]) could also promote bone colonization *in vivo*. Towards this end, we performed IIA injection of MCF-7 cells on animals without oestradiol supplement, which usually resulted in even more indolent bone colonization of this ER⁺ cell line. EPZ-6438 significantly increased bone colonization even under this oestradiol-deficient condition (Fig. 6c). Thus, it is likely that histone methylation may play an important role in cancer–niche interaction and resistance to oestrogen deprivation therapies. Because *EZH2* has been considered to be a tumour-promoting gene, we hypothesized that its function could be context-dependent in human metastases. We used an EZH2 target gene signature[29] as an index of EZH2 activities, and applied it to breast cancer metastases in different organs. Interestingly, a large variation was observed: bone and brain metastases express significantly lower levels of EZH2 target genes as compared to metastases in other organs (Fig. 6d). Thus,

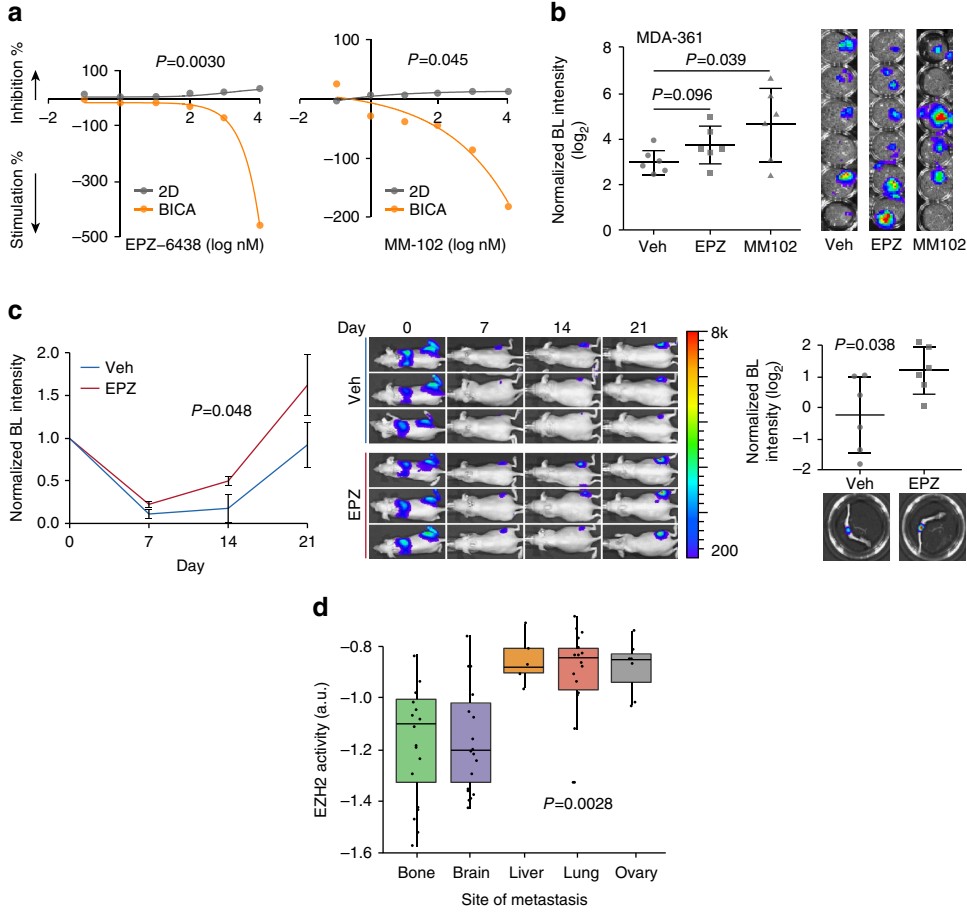

**Figure 6 | EPZ-6438 and MM-102 selectively stimulate the progression of bone micrometastases.** (**a**) Dose responsive curves of MCF-7 cells to EPZ-6438 (left) and MM-102 (right) in 2D culture and BICA. For 2D cultures, three technical replicates were included for each drug concentration. For BICA, $N = 5$ bone fragments for each drug concentration. Samples are under corresponding treatment for 3 weeks and measured by bioluminescence (BL). $P$ values are determined by repeated measures analysis of variance (ANOVA) tests. (**b**) The effects of EPZ-6438 and MM-102 (100 nM) on MDA-MB-361 cell in BICA. $N = 6$ bone fragments in each group. Samples are under corresponding treatment for 3 weeks. BL images are shown next to the dot plots. Veh, vehicle; EPZ, EPZ-6438. $P$ values versus control group are determined by least significant difference (LSD) test. Error bars: s.d. (**c**) The effects of EPZ-6438 on IIA-injected MCF-7 bone lesions. Left: growth curves as measured by *in vivo* BL imaging. $N = 6$ athymic nude mice per group. $P$ values are determined by repeated measures ANOVA tests on the growth curves. Error bars: s.e.m. Middle: representative BL pictures show bone lesion arising with or without EPZ-6438 treatment. Right: quantification of BL intensity after hindlimbs are extracted. Representative *ex vivo* BL imaging results are shown below the dot plot quantification. Error bars: s.d. The experiment was replicated twice and showed consistent results. Data shown are the combined results. The $P$ value was determined by $t$-test (two-tailed). (**d**) Box-whisker plots show the relative expression of EZH2 target genes in breast cancer metastases at different anatomical sites. The expression values are median-centred and log-transformed from the original microarray data set. Black dots superimposed on each box indicate individual specimen. Sample sizes are shown in parentheses. $P$ value is determined by one-way ANOVA.

the bone microenvironment may favour reduced EZH2 activities in cancer cells. These results demonstrate the utility of BICA for discovery of unexpected effects of compounds and their target pathways.

## Discussion

It has long been recognized that the microenvironment plays an important role in dictating metastatic progression and modulating therapeutic responses. Both direct cell–cell crosstalk and interaction with extracellular matrix can rewire signalling network inside cancer cells, thereby modulating their behaviours under specific conditions[30,31], probably through inducing epigenetic changes in cancer cells. Particularly in bone both osteoblasts and osteoclasts, as well as other resident cells, together constitute a unique cellular and molecular environment[5], which likely renders metastatic cells resistant to some and vulnerable to other perturbations as compared to the same cancer cells in other milieu. A prominent example is anti-osteoclast treatment with bisphosphonate and denosumab, which have been widely used in the clinic to limit the progression of osteolytic bone metastases[32]. More recently, several pathways have been shown to mediate cancer–bone crosstalk, including Notch[33], transforming growth factor-β (ref. 34), HIF[35,36], Integrin[37], Irf7 (ref. 38), VCAM1 (ref. 39) and mTOR[13]. These clinical and preclinical investigations highlighted the urgency of considering microenvironment when treating metastatic cancers. BICA meets this imperative need by recapitulating cancer–niche interaction in early-stage bone colonization and revealing bone-specific therapeutic responses.

Some previous studies also aimed to model cancer–bone interactions under ex vivo or in vitro settings[40]. One study added cancer cells to already-fragmented bone chips[41,42]. Others attempted to mimic one or a few aspects of the bone microenvironment by using extracellular matrix[43,44], hydrogel[45], man-made ceramic and composite scaffolds[46], or cell-derived matrices[47]. BICA differs significantly from these approaches, in that it delivers cancer cells into natural bone tissue via circulation, and allows the subsequent seeding and cancer–niche interaction to occur in vivo. Because of the tight cancer–bone adherence, several important features are maintained after fragmentation, including the cellular components of the metastasis niche, the gene expression of cancer cells, growth kinetics and the specific mechanisms of cancer–niche crosstalk (Figs 1–3). This faithfulness has not been demonstrated in any of the above ex vivo models.

Co-transplantation of human fetus bone fragments and cancer cells have been demonstrated. Spontaneous bone metastases occurred preferentially to human bones over mouse bones in this model. Moreover, the cancer–bone interaction is between human tissues[48]. However, restricted availability of fetus bone tissue limits the wide application of this approach. Moreover, it would be difficult to apply quantitative and rapid parallel assays with spontaneous metastasis models. BICA overcomes these barriers and provides a highly complementary platform to accelerate therapeutic and mechanistic studies of bone metastasis.

In the current study BICA was primarily used to model pre-osteolytic micrometastases, although we also demonstrated its potential to investigate the onset of osteoclast activation. Perhaps because of limited bone surface area and insufficient monocyte supply, the ability of BICA to model full-fledged vicious cycle is limited. The pre-osteolytic stage corresponds to the phase when adjuvant therapies are applied in the clinic. Although adjuvant therapies are intended to target micrometastases in distant organs, the decision and choice of specific treatments has to be made based on features of primary tumours. In this study we provided three specific examples of how cancer cells in a different context may respond differently to certain drugs. While danusertib exhibited enhanced efficacies on cancer cells interacting with bone, two HMT inhibitors unexpectedly showed the opposite trend—they promoted tumour growth, specifically in the bone microenvironment. Several types of mechanisms could lead to such markedly altered responses. First, cancer–bone interaction may rewire the signalling network in cancer cells, thereby altering their responses to certain drugs. Second, some drugs may act on the microenvironment niche cells, thereby indirectly affecting cancer cells. Third, biophysical and biochemical properties of the microenvironment niche may enrich or deprive certain drugs, thereby changing the bioavailability of these drugs to cancer cells. Future studies will be needed to delineate specific mechanisms behind each drug that exhibits distinct effects on cancer cells in the bone microenvironment.

We observed that endothelial cells could persist for weeks in BICA (Supplementary Fig. 1f). Moreover, we detected dormant cancer cells adjacent to endothelial cells (Supplementary Fig. 4d). These results support that the peri-vascular niche regulates metastasis dormancy[20,21]. Our previous studies demonstrated that the osteogenic niche promotes cancer cell proliferation[13]. Thus, cellular fates of cancer cells may be influenced by their distribution or migration between different microenvironment niches. More quantitative and, ideally, real-time imaging will be needed to further pursue this hypothesis. A limitation of BICA is the lack of immune cells other than monocytes. Adaptive immunity was shown to play an important role in bone metastasis[38]. This limitation may be partially overcome by reconstituting some of the immune cell components in the ex vivo environment of BICA in future studies.

## Methods

**Cell lines.** None of the cell lines used in this study are listed in the database of commonly misidentified cell lines maintained by ICLAC and NCBI Biosample. MCF-7, MDA-MB-231 and MDA-MB-361 were purchased from American Type Culture Collection. S.I. Abrams (National Cancer Institute) generously provided AT3 cells. For regular maintenance, cells were cultured in DMEM media supplemented with 10% fetal bovine serum (FBS). As the parallel control for BICA assay, 500 cells were seeded and then cultured in DMEM/F12 media supplemented with 2% FBS in 96-well plates. No cell line authentication was performed. Cell lines were subjected to bi-monthly tests for mycoplasma contamination.

**Lentivirus transduction of tumour cells.** The Fluc-GFP fusion gene was cloned into expression vector pwpt-GFP (Addgene #12255) in place of the GFP gene in the backbone. Using Xtreme Gene HP DNA Transfection Reagent (Version 08, Roche), the pwpt-Fluc-GFP vector was transfected into 293T cells with pMD2.G (Addgene #12259) and psPAX2 (Addgene #12260) to package lentivirus. Lentiviral stocks were filtered by 0.45 μm polyethersulfone membranes (VWR 28145–505). Cancer cells were incubated with Fluc-GFP lentivirus and 4 μg ml$^{-1}$ polybrene for 8 h. After a 72 h culture period, successfully labelled cells were isolated by FACS sorting of GFP-positive cells.

The inducible H2B-GFP system was reported[18] and used in our previous study. Briefly, the H2B-GFP fusion gene was introduced into pINDUICER22 vector[49] in place of the original GFP gene. Virus generation and transduction were performed using the same procedures as described above. A unit of 1 μg ml$^{-1}$ doxycycline treatment was performed for 3 days to induce the H2B-GFP protein expression in cells. The H2B-Fluc vector was modified from H2B-GFP system, which replaced the GFP gene with the Fluc gene.

**PDX model.** PDX 4664, gifted by Dr Mike Lewis' lab at Baylor College of Medicine, is an established PDX tumour model, which has been granted protocol exemption by the Institutional Review Board of Baylor College of Medicine for not involving human subjects.

**Immunofluorescent staining.** Immunofluorescent staining was performed as previously described[13]. Bone or BICA pieces were collected 3 weeks after tumour cell inoculation. Sample preparation was assisted by the shared Pathology Core resource of Dan L. Duncan Cancer Center. Staining was performed using antibodies against ALP (Abcam, ab108337, 1:500), Col-I (GeneTex, GTX41286,

1:500), CTSK (Abcam, ab19027, 1:100), Osterix (Abcam, ab22552, 1:500), CD31 (Abcam, ab28364, 1:100) and N-cadherin (BD, 610920, 1:250).

**Bone seeding model.** The assay was performed as previously described[42]. Femoral heads were fragmented and aligned into the low-attachment 96-well plates with 100 μl DMEM-F12 media with 2% FBS. MCF-7 cells were seeded on top of these slices at a density of 20,000 cells per well and then incubated and traced by bioluminescence imaging for 28 days.

**Extracellular matrix coating model.** The assay was performed as previously described, with slight modification[44]. Briefly, 500 cells were plated on top of solidified Matrigel Membrane Matrix (Corning, 354234) or Col-I (BD 354236) in each well of a 96-well plate. Cells were fed once every week with DMEM-F12 media with 2% FBS and traced by bioluminescence imaging for 28 days.

**3D suspension culture model.** A total of 20,000 luciferase-labelled cancer cells were cultured in a low-attachment 24-well plate in DMEM/F12 media with either 2% FBS or murine basic fibroblast growth factor (20 ng ml$^{-1}$, Life Technologies), as well as murine epidermal growth factor (20 ng ml$^{-1}$, Life Technologies) and B27 supplement (GIBCO). Cell growth was traced by bioluminescence imaging for 24 days.

**Animals.** Athymic nude mice and Balb/c mice were purchased from Harlan Sprague Dawley. *Osx1-GFP::Cre* mice, *Tie2-Cre* mice and *Cdh2$^{flox}$* mice were purchased from Jackson Laboratories (stock numbers: 006361; 004128; and 007611, respectively). These strains were intercrossed to produce experimental cohorts *Osx1-GFP::Cre/Cdh2$^{flox}$* and *Tie2-Cre/Cdh2$^{flox}$* mice. Mice were genotyped by PCR. To halt EGFP/Cre fusion protein expression, mice were provided with fresh drinking water containing 2 mg ml$^{-1}$ doxycycline (Sigma). The 5- to 7-week-old female mice were used in all *in vivo/ex vivo* experiments. The strain and number of mice used for each experiment are mentioned in text and legends. All animal work was done in accordance with a protocol approved by the Baylor College of Medicine Institutional Animal Care and Use Committee. The investigator was not blinded to the group allocation during the whole experiment.

**Injection and quantification of cancer cells in bone.** Intra-iliac injections, mammary pad injections and IVIS imaging were performed as previously described while the data normalization was processed with slight modifications[13]. After injection, animals were imaged weekly using IVIS Lumina II (Advanced Molecular Vision), following the manufacturer's recommended procedures and settings. For intra-iliac injection, we focused on bioluminescence signals at the epiphysis and metaphysis of femur and tibia other than the entire hindlimb. The acquired intensity data were divided over the day 0 signal intensity of the same animal at the same locus to yield normalized values. The intensity values were first normalized to day 0 and then subjected to statistical tests.

**Spontaneous bone metastasis assay.** The spontaneous bone metastasis assay was performed as previously described, with slight modifications[13]. Briefly, BALB/c mice were subjected to mammary fat pad injections of $1 \times 10^5$ 4T1.2 cells suspended in 50% Growth Factor Reduced Matrigel Matrix. Surgical resection was used to remove the tumours after 10–11 days. Mice were then randomized into different treatment groups and closely monitored for another 2 weeks. Bioluminescence imaging was performed before mice were killed. The Student's *t*-test was used to compute the *P* value.

**Pharmacological treatment.** Torin 1 (S2827, Selleckchem), PD98059 (A1663, ApexBio) and danusertib (A4116, ApexBio) were injected via intraperitoneal (i.p.) injection, daily, at the dosage of 6.8, 10 and 15 mg kg$^{-1}$, respectively. EPZ6438 (A8221, ApexBio) was injected via i.p. injection once every 2 days at 50 mg kg$^{-1}$. For antibody therapy, 4 mg kg$^{-1}$ DECMA-1 (U3254; Sigma) or rat immuno-globulin G (I4131; Sigma) was injected via i.p.injection twice a week. In the experiments using MCF-7 cells, oestradiol pellets were prepared and transplanted into nude mice before cancer cell injection, according to a previously published protocol[50], unless otherwise noted.

**Set-up of BICA.** Bone-in-culture pieces were prepared from the distal epiphysis and metaphysis of the femurs and the proximal epiphysis and metaphysis of the tibias collected from mice 30 min after receiving intra-iliac injection (Fig. 1a and Supplementary Fig. 1a). The bones were crushed with a sterilized bone plier (F.S.T. 16025–14), trimmed with micro dissecting scissor (Roboz RS-5912) and transferred to low-attachment 96-well plates by micro dissecting forceps (Roboz RS-5135) in a cell culture hood. Bone fragments derived from multiple animals were randomized and mixed in each group. Bone pieces bearing tumour cells were arranged in low-attachment 96-well plates and cultured in media comprises DMEM/F12. The concentration of FBS varied, with human at 2% and mouse at 0.5%. Medium was changed every 3–4 days, with medium being removed along the wall of wells

to avoid fragment aspiration. An IVIS Lumina II machine was used for bioluminescence imaging. The bioluminescence intensity data were divided over the day 0 signal intensity of the same piece to yield normalized values. Field of view = D; Exposure time = 1 min; bining = Medium; F/Stop = 1 for luminescent imaging. The intensity values were first normalized to day 0 and then subjected to statistical tests.

**Drug screening.** An epigenetics compound library used for drug screening was purchased from Selleckchem (Catalogue # L1900, 128cpds, ordered in June 2015). BICA samples preloaded with MCF-7 cells were aligned in low-attachment 96-well plates with six replicates per condition. In the first screening, 68 small-molecule epigenetic modulators were selected and grouped at a concentration of 100 nM for each compound based on information provided by the vender. In the secondary screenings, single compounds were tested at 100 nM separately. The names of selected compounds and group information are provided in Supplementary Table 1. For parallel experiments in 2D cultures, MCF-7 were plated in cell culture-treated 96-well plates at a density of 500 cells in 200 μl medium per well with six replicates per condition.

**RNA-seq experiment and data analysis.** All tumour samples were collected 3 weeks after inoculation. After homogenization, total RNA was extracted using the Direct-zol RNA miniprep kit. The first and second strands of cDNA were prepared by SuperScript III First-Strand Synthesis System and NEBNext mRNA Second Strand Synthesis Module from at least 200 ng total RNA, for each sample. Sequencing libraries were prepared from 1 ng of purified double-stranded cDNA with the Illumina Nextera XT DNA Library Prep Kit (Illumina, San Diego, CA, USA) according to the protocol supplied by the manufacturer. Cluster generation was performed using the Illumina Nextseq 500/550 high output v2 kit and sequenced on the Illumina Nextseq 500 equipment.

RNA-seq next-generation sequencing reads derived from xenograft materials containing a mixture of human and mouse reads were separated using Xenome (version 1.0.1)[51]. We used the default k-mer size suggested by Xenome (-k 25).

RNA-seq next-generation sequencing reads were mapped using STAR RNA-seq aligner (version 2.4.1d)[52]. To improve mapping accuracy, the gene transfer format file was supplied at the genome index generation step with the command line option—sjdbOverhang 79 (ReadLength - 1) together with the genome fasta file. To make full use of the reads not uniquely mapped we used RSEM[53], which applies estimation maximization to quantify gene and isoform expression. DEseq2 R package[54] was used to normalize the gene/isoform expression matrix.

Single sample gene set enrichment analysis (ssGSEA) was performed according to previous studies[55,56], which was used to calculate separate enrichment scores (ES) for each pairing of a sample and gene set in Molecular Signatures Database (MSigDB). ssGSEA (v7) analyses were performed using GenePattern (http://genepattern.broadinstitute.org/). This procedure is similar to GSEA. But unlike ranking the genes by comparing samples from two groups, the ssGSEA ranks the genes by their absolute expression in one sample. The ranked gene list is used to compute the ES with Empirical Cumulative Distribution Functions. And the $ES_{gene\ signature} = ES_{genes\ in\ the\ signature} - ES_{genes\ not\ in\ the\ signature}$.

The 13 bone marrow reference cells were downloaded from the E-MTAB-2923 data set[57]. Reads were mapped and quantified using STAR and were subsequently sorted using Samtools (version 0.1.19)[58]. HT-seq suite[59] was used. The reads count matrix was normalized using DEseq2. CIBERSORT[60] was used to estimate the relative percent of 13 bone marrow cell compositions. We used 1,000 permutation and disabled quantile normalization. EZH2 signature was downloaded from MSigDB http://software.broadinstitute.org/gsea/msigdb/collections.jsp and was computed as Σ Upregulated genes − Σ Downregulated genes.

**Statistical methodology.** All results are presented in the form of mean ± s.e.m., unless otherwise specified. Sample sizes for *in vivo* experiments are noted in the corresponding figures or figure legends. For *in vivo* experiments, sample size was determined by preliminary experiments or previous reports[13]. Randomization process was performed by randomly assigning animals into two separate groups after tumour inoculation, in Fig. 5e. No randomization was used in other experiments. No specific tests were conducted to test the assumption of normal distribution. Log-transformation were performed for most dot-plot graphs to achieve better visualization. Differences among growth curves and IC50 inhibition curves were assessed using repeated measure analysis of variance tests. In experiments consisting of more than two groups, the differences between means of different experimental groups were analysed using one-way analysis of variance tests, unless otherwise noted in respective legends. *F*-tests were performed to compare variations within different groups. Student's *t*-tests were performed with the assumption of equal variation. If the data did not meet the equal variation assumption, Welch correction or non-parametric analysis was performed.

**Code availability.** The R (ver 3.3.2) scripts used to generate data in Figs 1 and 3 and Supplementary Figs 1 and 3 are available in GitHub: https://github.com/lintian0616/bica.

**Data availability.** The raw data of RNA-seq results used for Fig. 1 and Supplementary Fig. 1 have been deposited to NIH Gene Expression Omnibus and assigned the accession number GSE84114. All other remaining data are available within the Article and its Supplementary Files, or available from the authors on request.

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

## Acknowledgements
We thank D. Weiss and Zhang Laboratory members for helpful input. We also thank Dr Abrams, S. for generously providing the AT3 cell line, and Dr Lewis, M.T. for providing the PDX4664 model. X.H.-F.Z. is supported by the US Department of Defense

DAMD W81XWH-16-1-0073 (Era of Hope Scholarship), NCI CA183878, Breast Cancer Research Foundation, Susan G. Komen CCR14298445 and McNair Medical Institute. H.W. is supported by the US Department of Defense DAMD W81XWH-13-1-0296. This work was also made possible by the CyVerse Collaborative, funded by the NSF (No. DBI-0735191). We also acknowledge the Integrated Microscopy Core at Baylor College of Medicine with funding from the NIH (HD007495, DK56338 and CA125123), the Pathology Core of Lester and Sue Smith Breast Center, the Dan L. Duncan Cancer Center and the John S. Dunn Gulf Coast Consortium for Chemical Genomics.

## Author contributions

X.H.-F.Z. and H.W. conceived the experiments and designed the research; H.W. and A.G. performed the experiments with assistance from H.-C.L., J.L., K.S., F.S., C.Z., Z.L., S.T.C.W., T.W. and M.A.M.; L.T. and H.W. analysed the data; X.H.-F.Z., H.W. and L.T. wrote the manuscript.

## Additional information

**Competing interests:** The authors declare no competing financial interests.

**Publisher's note**: 

