## [Peer Review File · Nature Communications]

Reviewers' comments:

Reviewer #1 (Remarks to the Author):

In their manuscript entitled "Bone-in-culture array (BICA) as a platform to model early-stage bone metastases and discover anti-metastasis therapies" Wang et al. introduce a new ex-vivo system to model the bone metastatic niche in breast cancer termed BICA. The authors employ gene expression profiling and drug screening to establish that this model reflects several aspects of the in-vivo context and contrast it with conventional two-dimensional cell culture. As a proof-of-principle the authors use BICA to predict that the pan-aurora kinase inhibitor Danusertib inhibits the progression of bone metastasis. They additionally uncover that histone methylation might have opposing effects on cancer cell growth in BICA versus conventional culture.

Given the overwhelming clinical need for novel therapeutic approaches for metastatic breast cancer the presented ex-vivo model has potential for high impact, as it provides a relevant system with higher-throughput capabilities than fully in-vivo models. The paper is well done and the studies are well described and highly innovative. This paper is well-suited for publication in Nature Communications. A few points from the paper might benefit from further supporting evidence and an extended discussion as outlined below. Additionally, experimental methods regarding the introduced model system should be extended to enable readers to fully recapitulate the proposed model.

Main points:

(i) The major novelty of the paper is the introduction of a new ex-vivo model for bone metastasis. In order to make the adoption of the model easier for readers further experimental details might be helpful. These could include how to select target bone areas, how to transfer fragments after crushing to ensure a comparable number of fragments per well, how to prevent fragment aspiration upon medium change, more details on bioluminescence measurements etc.

(ii) The authors highlight the importance of the microenvironment for metastatic growth and provide evidence that the interplay between bone and cancer cells is reflected in BICA. However, due to the lack of perfusion the model cannot recapitulate some of the impact of vasculature and the immune system on metastatic growth. Importantly, cancer and immune cell interactions could impact cancer dormancy. The authors might want to discuss these potential points regarding the use of the BICA to model metastatic progression and dormancy.

(iii) In Figure 5a the authors suggest that the bone microenvironment confers increased sensitivity of cancer cells to Aurora kinase inhibition, since administration of the Aurora kinase inhibitor Danusertib showed a 90% inhibition in BICA compared to 60% inhibition in two-dimensional culture. As shown in Figure 2b, especially early in cell culture cancer cells proliferate much faster in two-dimensional culture than in BICA. Do the authors believe it is possible that basal proliferation could account for the differences in treatment sensitivity rather than a microenvironment-specific effect on Aurora kinase signaling. In addition to showing inhibition rates it might thus be interesting to show data in the supplementary material of both growth kinetics with and without treatment rather than just inhibition ratios.

(iv) In Figure 5f-g the authors suggest that treatment with Danusertib eliminates dormant cancer cells using the inducible H2B-Fluc/GFP system. Staining directly for cancer cells could provide additional evidence that dormant cells are indeed removed. Since this would potentially be a very important clinical finding, the authors may wish (not required) to conduct a similar experiment by harvesting bones from an experiment analogous to the one shown in Fig 5d at a later timepoint and assessing for tumor cell persistence.

Minor points:

(i) The authors suggest that BICA can also be used to model the osteolytic phase of bone metastasis (Fig 1d). In these experiments were differentiation hormones such as M-CSF and RANKL supplemented?

(ii) Experimental protocols for viral transduction should be provided; additionally, the origin of the vectors used (e.g., H2B-Fluc) should be stated.

- (iii) The use of ssGSEAP (Fig 3a) should be mentioned in the method section.
- (iv) In Figure 3f-h the timepoint when bones were harvested should be indicated.
- (v) In Figure 3i-k it should be mentioned how long BICA and 2d cultures were treated for.
- (vi) Figures 3d-e: the time point of analysis should be stated.
- (vii) For Figure 3i-k and Fig 4b it should be stated, whether cell growth/inhibition was measured using bioluminescence.
- (viii) The time course for Fig 4b is not quite clear (duration of treatment).
- (ix) Statistics for the experiment in Fig 5g should be provided.
- (x) For statistics the authors state that for data with "large deviation" the data were first log-transformed and then statistical tests were applied. As such, it is not quite clear when data were analyzed in which way and whether data of a certain type (e.g., bioluminescence data) were analyzed consistently in the paper.

Reviewer #2 (Remarks to the Author):

In this paper, the authors establish an ex vivo bone metastasis model termed Bone-in-culture array (BICA) which could recapitulate cancer-bone interactions in pre-osteolytic and osteolytic phase based on authors' claims, and might be a good platform for bone metastatic drug screening. BICA is based on bone fragments containing cancer cells derived from hind limb bones immediately following intra-iliac tumor cell injection. Comparing to in vivo bone lesions (IVBL), BICA has similar tumor growth dynamics, gene expression profiles, similar tumor-osteoblastic interactions, and potentially can generate osteolytic bone degradation. As a proof-of-concept, the authors utilized BICA model as a potential drug discovery platform. Drug screening is performed and a few compounds with specific effects against bone metastatic on BICA were discovered. The strength of this manuscript includes a novel, very well developed ex vivo bone metastasis model, characters of which have been meticulously tested using multiple cell lines and pharmacologic agent perturbations (for N-E-cadherin junction and mTOR pathway). Besides, the authors also used a proof-of-concept screening demonstrating the difference of drug sensitivity on tumor cells in BICA model and in classical 2-D culture. With a few additional experimental data presentation and minor revision of the manuscript, this will be an excellent resource paper for the metastasis research field.

Specific comments:

1. The growth dynamics of bone tropic metastatic cell line in BICA and IVBL is not directly compared. It is known that many bone tropic metastatic breast cancer cell lines, include SCP28TR cells can expand up to thousand fold in mice based on BLI images in a matter of 6-8 weeks. It is thus desirable for authors to present BLI data from BICA and IVBL models in this time frame. This would be much more meaningful than presenting a few representative BLI images from BICA.
2. In Fig 1d and related supplementary figures, monocytes response to M-SCF/RANKL treatment and differentiate into TRAP+ osteoclasts, and TRAP+ cells will emerge at 2-3 weeks spontaneously (as osteolytic progress of tumor cells inferred by the authors). Without tumor cell injection, osteoclasts can often be found in normal bone structures. Besides, the progression of TRAP+ cells in 1D doesn't seem to be associated with increased tumor proliferation in that region. The authors should co-stain CTSK and GFP (tumor cells), this would be a better way to present tumor-associated osteoclast activity. Another related question is, at the very late stage (after 6 weeks of culture), will the authors be able to detect very strong bone destruction activity (bone mass decrease by CT) and massive TRAP+ staining on bones as seen by in vivo metastasis assay? In Supplemental Fig. 2C, it is clear, that osteoclast maturation is still a rare event and is not associated with tumor cell expansion. Thus, at least the current presented data doesn't support the claim that BICA can represent "osteolytic stages of bone colonization". Either a more complete data should be included or the text needs to be confined based on current data.

3. Even through both Ki67 and CC3 immunofluorescent stainings are negative in Fig 2d, lower magnification would be nice for readers to judge. Some quantitative experiments, such as the profiling of apoptotic or proliferative cells in GFP positive tumor cells by FACS, could be helpful.

4. Drug screening using BICA model was presented, three individual drugs were selected for further analysis. However, how these drugs cause tumor metastasis inhibition/acceleration through affecting tumor-bone stromal interactions were not clear and should be discussed.

Minor points:

1. Fig 1b, BICA model, DAPI staining is hardly detectable.

2. Many breast cancer cell lines (more than 5 lines) are used in this manuscript, which one is used in each experiment is not clearly marked. This information is especially missing in many figure legends. Please revise accordingly.

Reviewer #3 (Remarks to the Author):

In this well written manuscript the authors use the previously described experimental bone metastasis assay (Ref 13), fragment the bones after tumor cell inoculation and use these fragments to assay a number of parameters including drug activity while comparing these outcomes to those in 2D cultures. This fragmented bone assay is termed the "bone-in-culture array" or BICA. While superficially interesting, a deeper conceptual examination of this extensive experimental work comes up lacking for a novel contribution. There are several reasons for this conclusion:

1) Intravascular inoculation of cancer cells and observation of their migration to bone has been carried out for many years and indeed used in a number of drug discovery and development publications. Yet, even this in vivo assay does not have a good track record of predicting outcome of human clinical investigation and thus is not commonly used for such use. Hence, if this more sophisticated in vivo assay is not reliable predictor of patient effects, why should BICA be?

2) The work in this manuscript nicely shows that agents (drugs, cells etc...) examined in BICA behave differently than in 2D culture. Is this surprising? These are such vastly different conditions that it was a forgone conclusion this would be the case. Why not use 3D?

3) As the authors mention in the discussion, many other bone assays have been described. Several are easier to use and implement than BICA. However, the authors do not compare them to BICA and hence it is still unclear which is the best assay for bone metastasis. Without this assessment the contribution BICA offers to the bone metastasis field is unknown.

Reviewers' comments:

Reviewer #1 (Remarks to the Author):

In their manuscript entitled “Bone-in-culture array (BICA) as a platform to model early-stage bone metastases and discover anti-metastasis therapies” Wang et al. introduce a new ex-vivo system to model the bone metastatic niche in breast cancer termed BICA. The authors employ gene expression profiling and drug screening to establish that this model reflects several aspects of the in-vivo context and contrast it with conventional two-dimensional cell culture. As a proof-of-principle the authors use BICA to predict that the pan-aurora kinase inhibitor Danusertib inhibits the progression of bone metastasis. They additionally uncover that histone methylation might have opposing effects on cancer cell growth in BICA versus conventional culture. Given the overwhelming clinical need for novel therapeutic approaches for metastatic breast cancer the presented ex-vivo model has potential for high impact, as it provides a relevant system with higher-throughput capabilities than fully in-vivo models. The paper is well done and the studies are well described and highly innovative. This paper is well-suited for publication in Nature Communications. A few points from the paper might benefit from further supporting evidence and an extended discussion as outlined below. Additionally, experimental methods regarding the introduced model system should be extended to enable readers to fully recapitulate the proposed model.

Main points:

(i) The major novelty of the paper is the introduction of a new ex-vivo model for bone metastasis. In order to make the adoption of the model easier for readers further experimental details might be helpful. These could include how to select target bone areas, how to transfer fragments after crushing to ensure a comparable number of fragments per well, how to prevent fragment aspiration upon medium change, more details on bioluminescence measurements etc.

Re: We have revised Supplementary Fig. 1 and Methods, and added more details of BICA methodology. Please see highlighted parts in Method under sections “Animal studies” and “Setup of BICA” for this additional information.

(ii) The authors highlight the importance of the microenvironment for metastatic growth and provide evidence that the interplay between bone and cancer cells is reflected in BICA. However, due to the lack of perfusion the model cannot recapitulate some of the impact of vasculature and the immune system on metastatic growth. Importantly, cancer and immune cell interactions could impact cancer dormancy. The authors might want to discuss these potential points regarding the use of the BICA to model metastatic progression and dormancy.

Re: We agree with the reviewer that this is a major limitation of our model. For vasculature, we did observe that CD31+ endothelium could persist for up to 21 days in BICA (**Supplementary Fig. 1f**). Furthermore, we found some dormant cancer cells adjacent to endothelium (**Supplementary Fig. 4d**), which is consistent with previous studies in the literature. This new result suggests that with some modifications BICA may be useful to study perivascular niche in the future. For immune cells, we showed that monocytes remain close to the bone fragments

and retain the potential to differentiate into osteoclasts (**Fig. 1c** and **Supplementary Fig. 2b**). However, we do agree that our model would not be able to take into account other immune cells such as T cells and B cells. We have added a few sentences to the discussion to emphasize this limitation (see the last paragraph of Discussion).

(iii) In Figure 5a the authors suggest that the bone microenvironment confers increased sensitivity of cancer cells to Aurora kinase inhibition, since administration of the Aurora kinase inhibitor Danusertib showed a 90% inhibition in BICA compared to 60% inhibition in two-dimensional culture. As shown in Figure 2b, especially early in cell culture cancer cells proliferate much faster in two-dimensional culture than in BICA. Do the authors believe it is possible that basal proliferation could account for the differences in treatment sensitivity rather than a microenvironment-specific effect on Aurora kinase signaling.

Re: To provide some insights into this question, we sought to use a third platform in which cancer cells also proliferate slowly. We decided to employ the 3D suspension medium that is typically used for mammosphere assay. Cancer cells proliferate slowly (if at all) in this setting (**Supplementary Fig. 4a**). Despite the slow proliferation, they respond to Danusertib to a degree similar to that in 2D culture, but significantly different from that in BICA (**Supplementary Fig.7b**). Based on this data, we reasoned that the differential responses to Danusertib are unlikely due to different proliferation rates.

In addition to showing inhibition rates it might thus be interesting to show data in the supplementary material of both growth kinetics with and without treatment rather than just inhibition ratios.

Re: We have provided growth curves with or without 100 nM Danusertib in **Supplementary Fig. 7a**.

(iv) In Figure 5f-g the authors suggest that treatment with Danusertib eliminates dormant cancer cells using the inducible H2B-Fluc/GFP system. Staining directly for cancer cells could provide additional evidence that dormant cells are indeed removed.

Re: We have performed direct staining of cancer cells in **Supplementary Fig. 7f**.

Since this would potentially be a very important clinical finding, the authors may wish (not required) to conduct a similar experiment by harvesting bones from an experiment analogous to the one shown in Fig 5d at a later timepoint and assessing for tumor cell persistence.

Re: We have performed this experiment and provided the results in **Supplementary Fig. 7c**.

Minor points:

(i) The authors suggest that BICA can also be used to model the osteolytic phase of bone metastasis (Fig 1d). In these experiments were differentiation hormones such as M-CSF and RANKL supplemented?

Re: In **Fig. 1d**, no exogenous M-CSF or RANKL was added. We have clarified this in legend.

(ii) Experimental protocols for viral transduction should be provided; additionally, the origin of the vectors used (e.g., H2B-Fluc) should be stated.

Re: We have added this information to Methods under the section of “Lentivirus transduction of tumour cells”.

(iii) The use of ssGSEAP (Fig 3a) should be mentioned in the method section.

Re: We have added this information as a subsection titled “ssGSEA analysis” under the section of “RNA-seq Experiment and Data Analysis” in Methods.

(iv) In Figure 3f-h the timepoint when bones were harvested should be indicated.

Re: We have added this information to corresponding figure legends.

(v) In Figure 3i-k it should be mentioned how long BICA and 2d cultures were treated for.

Re: We have added this information to corresponding figure legends.

(vi) Figures 3d-e: the time point of analysis should be stated.

Re: We have added this information to corresponding figure legends.

(vii) For Figure 3i-k and Fig 4b it should be stated, whether cell growth/inhibition was measured using bioluminescence.

Re: We have added this information to corresponding figure legends.

(viii) The time course for Fig 4b is not quite clear (duration of treatment).

Re: We have added this information to corresponding figure legends.

(ix) Statistics for the experiment in Fig 5g should be provided.

Re: Information updated in figure 5g.

(x) For statistics the authors state that for data with “large deviation” the data were first log-transformed and then statistical tests were applied. As such, it is not quite clear when data were analyzed in which way and whether data of a certain type (e.g., bioluminescence data) were analyzed consistently in the paper.

Re: In the revised version of the manuscript, we keep data analysis and presentation consistently for bioluminescence data. The vast majority of the growth curves were drawn on linear scales while the dot-plot graphs were drawn with log-scale for better visualization of intergroup differences unless specifically stated.

For statistical analysis, F-tests were performed to compare variations within different groups first. Student’s t-tests were performed with the assumption of equal variation. If the data did not meet the equal variation assumption, Welch correction or non-parametric analysis was performed.

This information is also included into the manuscript

Reviewer #2 (Remarks to the Author):

In this paper, the authors establish an ex vivo bone metastasis model termed Bone-in-culture array (BICA) which could recapitulate cancer-bone interactions in pre-osteolytic and osteolytic phase based on authors' claims, and might be a good platform for bone metastatic drug screening. BICA is based on bone fragments containing cancer cells derived from hind limb bones immediately following intra-iliac tumor cell injection. Comparing to in vivo bone lesions (IVBL), BICA has similar tumor growth dynamics, gene expression profiles, similar tumor-osteoblastic interactions, and potentially can generate osteolytic bone degradation. As a proof-of-concept, the authors utilized BICA model as a potential drug discovery platform. Drug screening is performed and a few compounds with specific effects against bone metastatic on BICA were discovered. The strength of this manuscript includes a novel, very well developed ex vivo bone metastasis model, characters of which have been meticulously tested using multiple cell lines and pharmacologic agent perturbations (for N-E-cadherin junction and mTOR pathway). Besides, the authors also used a proof-of-concept screening demonstrating the difference of drug sensitivity on tumor cells in BICA model and in classical 2-D culture. With a few additional experimental data presentation and minor revision of the manuscript, this will be an excellent resource paper for the metastasis research field.

Specific comments:

1. The growth dynamics of bone tropic metastatic cell line in BICA and IVBL is not directly compared. It is known that many bone tropic metastatic breast cancer cell lines, include SCP28TR cells can expand up to thousand fold in mice based on BLI images in a matter of 6-8 weeks. It is thus desirable for authors to present BLI data from BICA and IVBL models in this time frame. This would be much more meaningful than presenting a few representative BLI images from BICA.

Re: In **Supplementary Fig. 2f** we provided direct comparison between SCP28 cells in BICA and in IVBL in the time frame of 1-5 weeks. After 4 weeks, cancer cell growth in BICA began to lag behind probably due to limited bone surface area in fragments. At Week 5, BICA exhibited increased TRAP staining, similar to IVBL, suggesting onset of osteoclast activation. However, we agree with the reviewer that BICA has limited capacity in modeling late-stage bone metastasis. We have clarified this in the main text (see main text p.5 and discussion p.12).

2. In Fig 1d and related supplementary figures, monocytes response to M-SCF/RANKL treatment and differentiate into TRAP+ osteoclasts, and TRAP+ cells will emerge at 2-3 weeks spontaneously (as osteolytic progress of tumor cells inferred by the authors). Without tumor cell injection, osteoclasts can often be found in normal bone structures. Besides, the progression of TRAP+ cells in 1D doesn't seem to be associated with increased tumor proliferation in that region. The authors should co-stain CTSK and GFP (tumor cells), this would be a better way to present tumor-associated osteoclast activity.

Re: We have revised **Fig. 1d** by co-staining CTSK and GFP. We observed that large size lesions in BICA were associated with stronger osteoclast activity. This association is now

quantitated in **Supplementary Fig. 2c**. Consistent with the growth curves discussed in the previous point, this data suggests that BICA recapitulates osteoclast activation to some degree. However, probably due to limitation of bone surface area and monocyte supply, BICA may not be able to model full-fledged vicious cycle. Again, this has been discussed now in the main text.

Another related question is, at the very late stage (after 6 weeks of culture), will the authors be able to detect very strong bone destruction activity (bone mass decrease by CT) and massive TRAP+ staining on bones as seen by in vivo metastasis assay? In Supplemental Fig. 2C, it is clear, that osteoclast maturation is still a rare event and is not associated with tumor cell expansion. Thus, at least the current presented data doesn't support the claim that BICA can represent "osteolytic stages of bone colonization". Either a more complete data should be included or the text needs to be confined based on current data.

Re: In BICA it is difficult to evaluate bone destruction by mass or volume, as the specimens are bone fragments, not intact bones. However, another index, "surface to volume ratio" is a useful measurement because bone dissolving is expected to result in rough bone surface and an increased surface-to-volume ratio (see micro-CT below: A=normal bone fragment, B=tumor-containing bone fragment in BICA). We have included a quantitation of these results as **Supplementary Fig. 2d**.

3. Even through both Ki67 and CC3 immunofluorescent stainings are negative in Fig 2d, lower magnification would be nice for readers to judge.

Re: We now include lower magnification pictures in **Fig 2d**. Of note, during the revision we found that CC3 is not an ideal apoptosis marker for MCF-7 (Oncogene (2001) 20, 6570 - 6578). So we used BAX to replace CC3 in the revision.

Some quantitative experiments, such as the profiling of apoptotic or proliferative cells in GFP positive tumor cells by FACS, could be helpful.

Re: Once cancer cells colonize in the bone, it was difficult to extract all of them. A major population of them got embedded into the bone matrix, and cannot be easily dissociated. We have published this recently (J Mammary Gland Biol Neoplasia (2015) 20:103–108). As a result, we have been performing quantification based on immunofluorescence staining.

4. Drug screening using BICA model was presented, three individual drugs were selected for further analysis. However, how these drugs cause tumor metastasis inhibition/acceleration through affecting tumor-bone stromal interactions were not clear and should be discussed.

Re: We have added some speculated mechanisms in discussion (see the second last paragraph in Discussion). We agree with the reviewer that the underlying mechanisms are interesting, and will be elucidated in our future studies.

Minor points:

1. Fig 1b, BICA model, DAPI staining is hardly detectable.

Re: We have increased the exposure time of DAPI channel, and the signal is now clear.

2. Many breast cancer cell lines (more than 5 lines) are used in this manuscript, which one is used in each experiment is not clearly marked. This information is especially missing in many figure legends. Please revise accordingly.

Re: We have added this information to figure legends.

Reviewer #3 (Remarks to the Author):

In this well written manuscript the authors use the previously described experimental bone metastasis assay (Ref 13), fragment the bones after tumor cell inoculation and use these fragments to assay a number of parameters including drug activity while comparing these outcomes to those in 2D cultures. This fragmented bone assay is termed the “bone-in-culture array” or BICA. While superficially interesting, a deeper conceptual examination of this extensive experimental work comes up lacking for a novel contribution. There are several reasons for this conclusion:

1) Intravascular inoculation of cancer cells and observation of their migration to bone has been carried out for many years and indeed used in a number of drug discovery and development publications. Yet, even this in vivo assay does not have a good track record of predicting outcome of human clinical investigation and thus is not commonly used for such use. Hence, if this more sophisticated in vivo assay is not reliable predictor of patient effects, why should BICA be?

Re: We respectfully disagree with the reviewer’s dismissal of all animal model-based research on bone metastasis in general. Bone metastasis is arguably the best understood metastasis, and is the only metastasis with specific treatments available in the clinic. Bisphosphonates and denosumab have both exhibited remarkable efficacies in controlling metastatic progression in the bone. Although bone metastasis remains incurable, these drugs significantly improve the quality of life of patients. In fact, they were developed exactly because of earlier work by Greg Mundy and many others in 80’s, which elucidated the osteolytic vicious cycle in inoculation-based animal models.

That said, we understand the reviewer’s concern that intravascular inoculation of cancer cells bypasses earlier steps of metastasis. This is an important caveat, which we have attempted to overcome using a spontaneous bone metastasis model (see **Fig. 5e**). The advantage of BICA is that it may achieve a balance between biological sophistication and experimental scalability: it recapitulates key biological properties of metastasis while allowing fast pre-clinical tests. As can be seen in our manuscript, we always attempted to validate findings from BICA using in vivo models or even clinical evidence. What we hope is that BICA will accelerate screening of biological agents against bone metastasis in pre-clinical settings.

2) The work in this manuscript nicely shows that agents (drugs, cells etc...) examined in BICA behave differently than in 2D culture. Is this surprising? These are such vastly different conditions that it was a forgone conclusion this would be the case.

Re: We agree that it is easy to predict that there must be some differences between BICA and 2D culture regarding therapeutic responses. However, based on our current knowledge it is impossible to predict on which drugs we should expect to see such differences. It is our hypothesis that cancer-bone interaction may alter the signaling network in cancer cells and render them resistant to some drugs but vulnerable to some others. Yet, we cannot predict the identity of these drugs and the direction of differences without doing the BICA experiments. A

striking example is the histone methyl-transferase inhibitors, drugs that are under clinical development to cure cancer, were actually promoting tumor growth specifically in the bone microenvironment. Learning about these unexpected effects of drugs in our opinion, is an important step toward biological mechanisms and clinical applications.

Why not use 3D?

Re: In this revision, we have included a 3D condition, and showed that cancer cells in this condition exhibited therapeutic responses distinct from BICA and IVBL (**Supplementary Fig. 7b**).

3) As the authors mention in the discussion, many other bone assays have been described. Several are easier to use and implement than BICA. However, the authors do not compare them to BICA and hence it is still unclear which is the best assay for bone metastasis. Without this assessment the contribution BICA offers to the bone metastasis field is unknown.

Re: Indeed, some of the methods involve sophisticated engineering of biomaterials, which, in our opinion, are not nearly as feasible as our approach. Thus, we focused on a few other approaches that are easy to implement in an average laboratory. These include 3D culture, ECM-coated plates, and an approach directly dropping cancer cells on to bone fragments. These results are shown in **Supplementary Fig. 4a-c**. As can be seen, tumor growth kinetics are dramatically different under these conditions as compared to BICA and in vivo bone lesions.

REVIEWERS' COMMENTS:

Reviewer #1 (Remarks to the Author):

The authors have addressed my comments. I strongly support publication of this manuscript in Nature Communications. This is a very well done study by a pioneering laboratory and will make a major impact on the breast cancer field.

Reviewer #2 (Remarks to the Author):

The authors have carefully revised the manuscript in response to the reviewers' comments and the manuscript is now acceptable for publication.

Editor note:

Reviewer 3 was not available at this time. Therefore we consulted with reviewer 2 who commented for the editors only and was supportive of publication.

REVIEWERS' COMMENTS:

Reviewer #1 (Remarks to the Author):

The authors have addressed my comments. I strongly support publication of this manuscript in Nature Communications. This is a very well done study by a pioneering laboratory and will make a major impact on the breast cancer field.

Re: We appreciate the reviewer's comment.

Reviewer #2 (Remarks to the Author):

The authors have carefully revised the manuscript in response to the reviewers' comments and the manuscript is now acceptable for publication.

Re: We thank the reviewer's support.